

# RIP-MD: a tool to study residue interaction networks in protein molecular dynamics

Sebastián Contreras-Riquelme[1,2,3], Jose-Antonio Garate[4],
Tomas Perez-Acle[1,4] and Alberto J.M. Martin[3]

[1] Computational Biology Laboratory (DLab), Fundacion Ciencia & Vida, Santiago, Chile
[2] Facultad de Ciencias de la Vida, Universidad Andrés Bello, Santiago, Chile
[3] Network Biology Laboratory, Centro de Genómica y Bioinformática, Facultad de Ciencias, Universidad Mayor, Santiago, Chile
[4] Centro Interdisciplinario de Neurociencia de Valparaíso, Valparaíso, Chile

Corresponding authors
Tomas Perez-Acle, tomas@dlab.cl
Alberto J.M. Martin,
alberto.martin@umayor.cl

## ABSTRACT

Protein structure is not static; residues undergo conformational rearrangements and, in doing so, create, stabilize or break non-covalent interactions. Molecular dynamics (MD) is a technique used to simulate these movements with atomic resolution. However, given the data-intensive nature of the technique, gathering relevant information from MD simulations is a complex and time consuming process requiring several computational tools to perform these analyses. Among different approaches, the study of residue interaction networks (RINs) has proven to facilitate the study of protein structures. In a RIN, nodes represent amino-acid residues and the connections between them depict non-covalent interactions. Here, we describe residue interaction networks in protein molecular dynamics (RIP-MD), a visual molecular dynamics (VMD) plugin to facilitate the study of RINs using trajectories obtained from MD simulations of proteins. Our software generates RINs from MD trajectory files. The non-covalent interactions defined by RIP-MD include H-bonds, salt bridges, VdWs, cation-$\pi$, $\pi$–$\pi$, Arginine–Arginine, and Coulomb interactions. In addition, RIP-MD also computes interactions based on distances between $C_\alpha$s and disulfide bridges. The results of the analysis are shown in an user friendly interface. Moreover, the user can take advantage of the VMD visualization capacities, whereby through some effortless steps, it is possible to select and visualize interactions described for a single, several or all residues in a MD trajectory. Network and descriptive table files are also generated, allowing their further study in other specialized platforms. Our method was written in python in a parallelized fashion. This characteristic allows the analysis of large systems impossible to handle otherwise. RIP-MD is available at http://www.dlab.cl/ripmd.

# INTRODUCTION

The function of proteins is determined by both their 3D structure and their behavior. Therefore, the traditional dogma *sequence → structure → function* is currently restated as

*sequence→structure→dynamics→function*. Not surprisingly, molecular dynamic (MD) methods have become essential tools to explore protein dynamics with atomic resolution (*Van Gunsteren et al., 1995*). Notwithstanding, one of the major caveats of MD is the huge amount of data produced by each simulation, hindering both data handling and analysis. Thus, the generation of proper tools to conduct structural and dynamic analyses, is an area of active research and development (*Hub & De Groot, 2009*; *Hayward & De Groot, 2008*). Among other methods, the production of residue interaction network (RINs) gained popularity due to their simple and intuitive approach. RINs are a graph representation of protein structures in which nodes represent amino acid (AAs) and the existence of an interaction between two AAs is represented by edges. Some interesting usage of RINs in MD include the identification of key AAs involved in several processes such as allosterism (*Sethi et al., 2009*; *Malod-Dognin & Pržulj, 2014*; *Kaur Grewal, Mitra & Roy, 2015*), enzymatic activity (*Jianhong Zhou, Hu & Shen, 2016*), protein folding (*Vendruscolo et al., 2002*) and in protein–protein interactions (*Del Sol & O'Meara, 2005*). RINs have also been adopted for the prediction of the effect of single point mutations in protein stability (*Giollo et al., 2014*); the analysis of protein stability (*Brinda & Vishveshwara, 2005*); the superimposition and comparison of protein structures (*Gupta, Mangal & Biswas, 2005*; *Malod-Dognin & Pržulj, 2014*); assessing the quality of predicted protein structures (*Tress & Valencia, 2010*); and the study of light dark transitions in photoreceptors (*Kaur Grewal, Mitra & Roy, 2015*).

Many types of RINs can be defined depending on how the interactions between AAs are outlined. The most common definition is the Euclidean distance between atoms belonging to different residues, and thus interactions exists only if the distance between AAs is shorter than a given threshold. When this distance is calculated between $C_\alpha$s or $C_\beta$s, RINs are traditionally known as contact map (*Sethi et al., 2009*; *Brown et al., 2017*). RINs based on thresholded distances are simple graphs, that is, a pair of nodes can only be connected by a single edge. Nevertheless, there is another approach to build RINs that considers explicitly all non-covalent interactions pertaining a pair of AAs (*Martin et al., 2011*; *Wolek, Gómez-Sicilia & Cieplak, 2015*), giving rise to a multigraph in which nodes can be connected by several edges. In this way, each edge represents a single non-covalent interaction. Non-covalent interactions depicted by edges usually include salt bridges (SBs), hydrogen bonds (HBs), π–π, cation-π and van der Waals (vdW) contacts among others. Multigraph RINs contain more information than simple graphs built upon distance thresholds, since all interactions are described in the graph. As a drawback, the existence of several connections between pairs of nodes prevents the use of common algorithms and metrics that can be applied to characterize and analyze simple graphs (see *Brandes (2008)* for a discussion of the calculation of shortest paths in multigraphs).

Most of the current approaches to depict protein structures as a network are static (*Yan et al., 2014*), representing a single protein structure as a RIN disregarding the dynamic properties of the AAs and atoms forming it. Lately, several authors have introduced RIN based methodologies on full MD trajectories, the so-called dynamic RINs, allowing for the generation of consensus graphs that characterizes and contains the (averaged) dynamic properties of each residue with respect to the rest

(*Bhattacharyya, Bhat & Vishveshwara, 2013*; *Eargle & Luthey-Schulten, 2012*; *Münz & Biggin, 2012*; *Pasi et al., 2012*). Nonetheless, all the aforementioned methods are based on distance thresholds between single atoms lacking the important information provided by the explicit inclusion of non-covalent interactions.

Herein, we present residue interaction networks in protein molecular dynamics (RIP-MD), a software to generate both static and dynamics RINs. RIP-MD is available as a web server for RIN derivation using static protein structures (PDB files (*Berman et al., 2000*)), or as a visual molecular dynamics (VMD) (*Humphrey, Dalke & Schulten, 1996*) plugin to obtain dynamic RINs derived from MD trajectories. RIP-MD is meant to be an intuitive and easy-to-follow visualization tool for (pairwise) residue interactions. The latter is particularly relevant for MD in which the generated data is enormous. However, due to the employment of networks, the users can take advantage of the mathematical formalism of network theory (i.e., by using Cytoscape) to further analyze their simulations. RIP-MD reads MD trajectory files in DCD binary format generating snapshots at fixed time intervals from which a dynamic RIN is built on. RIP-MD can be employed to perform several analyses, including cross-correlation studies between the different types of interactions; generation of a consensus RIN where edges exists if they are present in at least a given percentage of the snapshots; and the calculation of different node centralities. In the next sections a detailed description of RIP-MD is elaborated. Furthermore, the utility of RIP-MD is emphasized with two examples of MD trajectories: the analysis of possible interactions that stabilize the structure of a gap-junction channel and the structural changes on the Lymphocyte antigen 96 protein.

A web server for the creation of RINs from single PDB files, the stand alone and the VMD plug-in versions of RIP-MD can be accessed and downloaded from dlab.cl/ripmd. In addition, to facilitate the use of RIP-MD under any operative system we also provide it already installed in a VirtualBox® machine.

## METHODS

The general workflow of RIP-MD is summarized in Fig. 1. RIP-MD starts either with a dynamic (MD trajectory) or static (PDB file) protein structure, and the parameters defining the interactions as input (Fig. 1A). The next step pre-processes the input to ensure it complies with the required format (Fig. 1B). Then, a search for interactions between all atoms is carried out in each snapshot from the MD trajectory or using the static structure from the PDB file (see Fig. 1C). In the last step (Fig. 1D), RIP-MD generates the output files including correlation maps and files defining a RIN, which can be further characterized in network visualization tools such as Cytoscape (*Shannon et al., 2003*). Each of these steps is described in detail below.

### Input and pre-processing steps

As previously mentioned, RIP-MD takes as input either a dynamic or a static protein structure (Fig. 1A). Structural analyses within RIP-MD are handled by MD analysis (*Michaud-Agrawal et al., 2011*; *Gowers et al., 2016*), a python library to manipulate MD trajectories. The first action in this step is to delete heteroatoms maintaining only

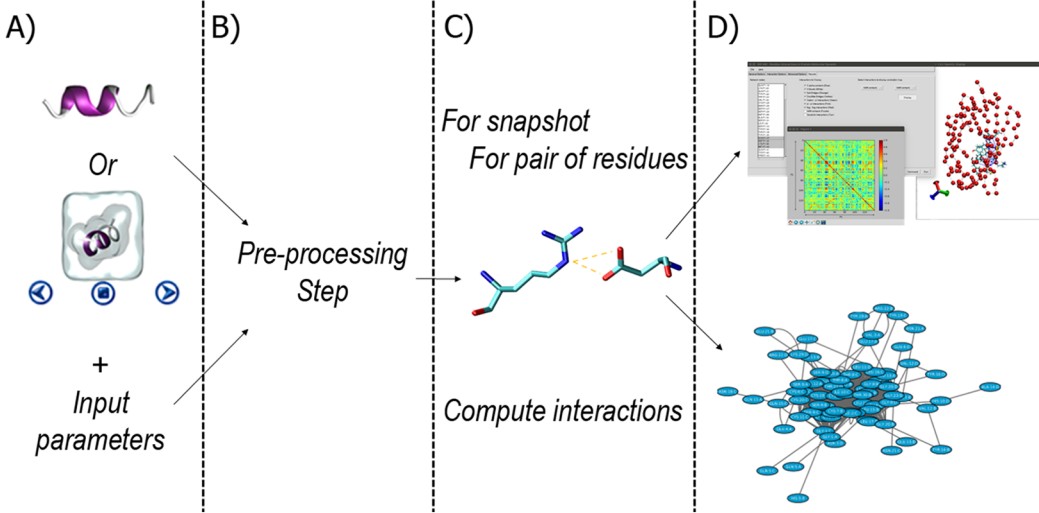

**Figure 1 Workflow in RIP-MD.** (A) input of structural information and analyses parameters. (B) Pre-processing step. (C) Definition of interactions ($C_\alpha$ contacts, H-bonds, Salt bridges, disulfide bonds, cation-$\pi$, $\pi$–$\pi$, Arginine–Arginine, Coulomb, and van der Waals contacts) according to the input parameters. (D) Generation of RIN and output files.

**Table 1 Summary of all interactions defined in RIP-MD.**

| | | |
|---|---|---|
| $C_\alpha$ contacts | dist ($C_{\alpha i}$, $C_{\alpha j}$) $\leq d$ | $d = 8$ Å |
| Hydrogen bonds | *dist* (donor, acceptor) $\leq d$ | $d = 3$ Å |
| | $\theta(\overrightarrow{C-H}, \overline{\text{acceptor}}) \geq a$ | $a = 120°$ |
| Salt bridges | Contacts between NH/NZ groups of ARG/LYS and OE*/OD* in ASP/GLU $\leq d$ | $d \leq 6$ Å |
| Disulfide bonds | S atoms of two cysteins $\leq d$ dihedral | $d \leq 3$ Å |
| | $\theta(C-S-S-C) \in [a, b]$ | $a \geq 60°$ |
| | | and $b \leq 90°$ |
| Cation-$\pi$ interactions* | Distance between aromatic rings $\leq d$ | $d \leq 6$ Å |
| $\pi$–$\pi$ interactions* | dist (aromatic ring, cation) $\leq d$ | $d = 7$ Å |
| | $\theta$ (normal vector ring, $\overrightarrow{\text{ringcenter} - \text{cation}}$) $= a$ | $a \in [0°, 60°]$ |
| | | or $a \in [120°, 180°]$ |
| Arg–Arg | dist (guanidine$_1$, guanidine$_2$) $\leq d$ | $d \leq 5$ Å |
| Coulomb interactions | Charged-group based cut-off using a 1–4 potential | |
| van der Waals | 12-6 Lennard–Jones potential | |

Notes:
A detailed description can be found in Text S1.
\* His residues are considered as a cation if they present a protonated nitrogen atom, and as $\pi$-system only if they are not protonated. Other $\pi$-systems considered are the aromatic rings of Phe, Tyr, and Trp.

protein atoms. If the user wants to add missing atoms, such as hydrogens, these are added based on internal coordinates with PDB2PQR (*Dolinsky et al., 2007*, *2004*). After all these steps, additional parameters such as (partial) charges, Lennard–Jones parameters and secondary structure are either defined or calculated. In the case of MD trajectories, partial charges are assigned from the topology file, that is, PSF file, employed to perform the simulation. Charges for static structures and Lennard–Jones parameters are assigned from the parameters files of the CHARMM force field (*MacKerell et al., 1998*).
Secondary structure and solvent accessibility for each AA are defined using DSSP (*Joosten et al., 2011*; *Kabsch & Sander, 1983*). For further details about input files and their format, please refer to the RIP-MD user manual.

## Interactions defined in RIP-MD

Residue interaction networks in protein molecular dynamics defines several types of interactions between the AAs in a protein structure: $C_\alpha$ contacts, HBs, SBs, disulfide bonds, cation-π, π–π, Arg–Arg, Coulomb, and vdW contacts. Parameters defining each interaction, together with their mathematical formulation is explained in detail in Text S1 and resumed in Table 1. Importantly, users may employ the provided by-default parameters defining each of these interactions or set them accordingly to his/her needs.

## RIP-MD versions and availability

Residue interaction networks in protein molecular dynamics is available free of charge at http://www.dlab.cl/ripmd in three options:

- Standalone program: This version is thought for users who want to take advantage of high-performance computing architectures to perform analysis of very large systems, such very long MD trajectories that are impossible to handle otherwise.
- Visual molecular dynamics (*Humphrey, Dalke & Schulten, 1996*) plugin: This version benefits from the graphical interface provided by VMD. This plugin performs a system call to execute the standalone RIP-MD program.
- Webserver: This form of RIP-MD is for those users who want to analyze a single PDB structure without installing the software locally. In this web-server, users first upload their selected structure and then, after few steps, the results can be easily downloaded.

It is important to note that results generated by the web-server and by the stand-alone version of the program are compatible with the VMD plugin, making possible to load and display these results into the VMD graphical user interface. The installation guide for both the stand-alone and plug-in versions, together with the user manual, is available at the RIP-MD webpage. In addition, users can also download from the webpage a virtual machine with RIP-MD already available which only requires the installation of all proprietary software that cannot be distributed by us.

Other scripts employed to generate the figures shown in this article are also available at http://www.dlab.cl/ripmd or directly at https://github.com/networkbiolab/supp_script_ripmd.

## Output files

Residue interaction networks in protein molecular dynamics provides two types of output: interaction graphs and Pearson correlation plots.

### Residue interaction networks

Once the interactions are computed, RIP-MD generates network files that can be visualized in specialized platforms such as Cytoscape (*Shannon et al., 2003*). In these networks, each node represents an AA and each edge represents an interaction between

AAs. Several network files are generated; one for each type of interaction type and a global network containing all interactions. Network files also describe additional information about several properties of the AAs including secondary structure and solvent accessible surface. These network files also contain edge attributes and the geometrical descriptors defining each interaction. To account for the dynamic behavior of node descriptors and interactions, RIP-MD calculates the percentage of frames in which they are present over the simulation.

### Pearson correlation plots

Residue interaction networks in protein molecular dynamics employs Pearson correlation to identify, for any given type of interaction, the existence of relationships between the dynamic behavior of two AAs. To do so, each AA is represented by a numerical vector in which each element contains the number of interactions of a given type in each frame. Once all vectors are obtained, the existence of a relationship between the rupture or formation of interactions over time is determined by calculating the Pearson correlation between these vectors. These results are provided in the form of correlation plots, square matrices of size $N$ representing the total number of AAs, where each element is colored according to the correlation value.

## Comparison with other methods

A comparison with two other approaches, Carma (*Glykos, 2006*) and MD-TASK (*Brown et al., 2017*), used on our first example (the MD2 pocket closure, see next section) is explained in detail in Text S1. These methods were chosen due to their availability, they are easy to install and they consider both inter and intra chain interactions, even thought both methods only generate a single type of interactions between residues ($C_\alpha$ and $C_\beta$ contacts for Carma and MD-TASK, respectively). Interested readers can also find in Text S1 a brief description of other methods or libraries that can be used to generate RINs for MD simulations, even if, as Carma and MD-TASK, they use MDs instead of non-covalent interactions in their RIN definitions.

## APPLICATION OF RIP-MD TO ANALYZE MD SIMULATIONS OF PROTEINS

In the following section, we explore two applications of RIP-MD to analyze MD simulations of proteins: conformational changes occurring in a soluble protein and the inter-monomeric interactions of a gap-junction channel (Fig. 2). Both analysis were carried out employing the by-default parameters of RIP-MD. The first example (Figs. 3 and 4) is meant to show how RIP-MD can visually discriminate among the different stages of a conformational change and how correlated interactions guide the process and are lost when the "closed" state is reached, which might be a common trend in process guided by non-polar interactions. Our new analysis demonstrates new chemical information that was not reported in (*Garate & Oostenbrink, 2013*), the original studies in which for the first time the closing process was reported. The second example is meant to emphasize how interactions in X-ray protein crystals are not necessarily the same.

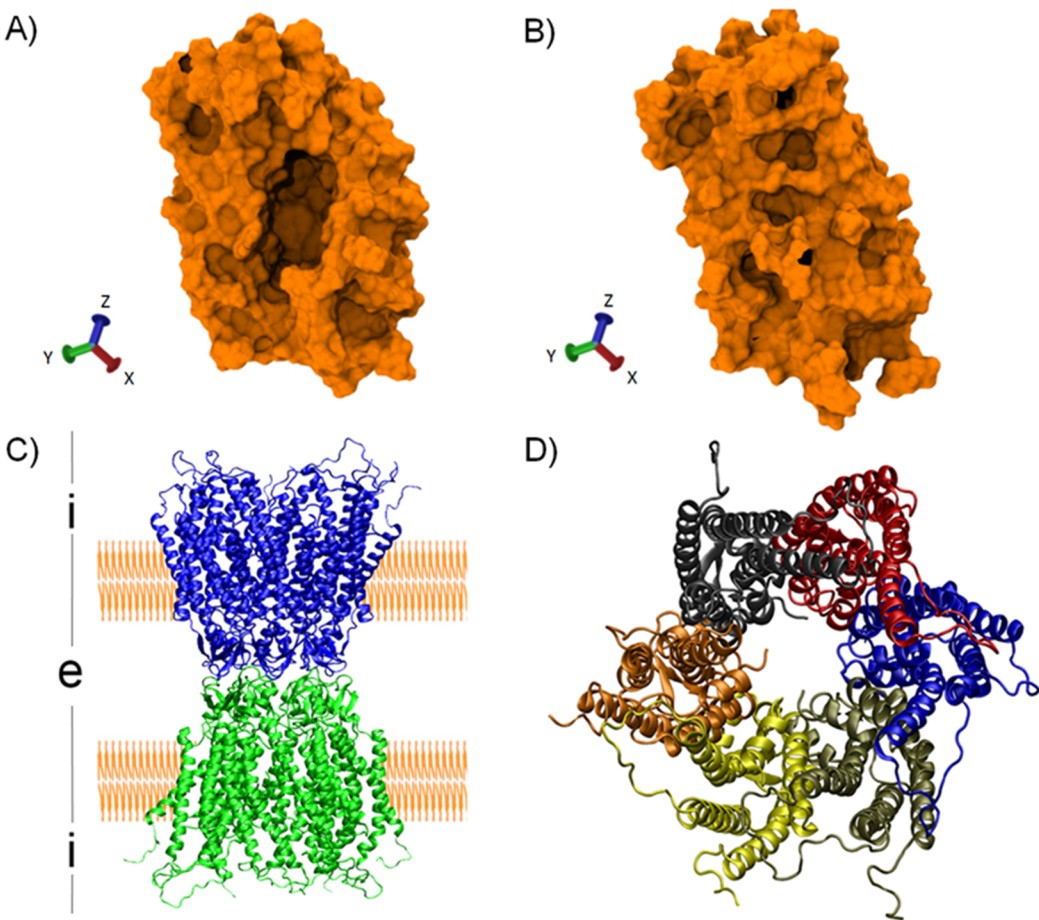

**Figure 2 Protein structures used as case of study.** First and last snapshots of the MD2 trajectory simulation (top) and molecular structure of the human CX26 hemichannel and gap-junction channel (bottom). (A) First snapshot of the MD showing the hydrophobic pocket in an open conformation. (B) Last snapshot of the MD where MD2 exhibit a closed conformation. (C) Secondary structure representation of a gap junction channels (GJC) formed by the extracellular docking of two HCs (blue and green). The position of the plasmatic membrane appears rendered in orange. The extracellular space is denoted with an "i", while the intracellular space is denoted with an "e". (D) Secondary structure representation of a HC from the intracellular view. Each CX26 monomer is represented using different colors.                                                  

Moreover, the visualization power of RIP-MD is highlighted as a very complex system (12 monomers) is very easily visualized by our method.

## MD2: an example of a conformational change

MD2 is a soluble protein part of the innate immune response in humans. Upon binding of lipopolysaccharides belonging to Gram-bacteria, MD2 triggers an immune response via the interaction with Toll-Like receptors 4 (*Dziarski et al., 2001*; *Ohto et al., 2007*). As shown by MD simulations (*DeMarco & Woods, 2011*; *Garate & Oostenbrink, 2013*), MD2 possesses a hydrophobic cavity that rapidly closes upon ligand removal (Fig. 2). The later is a clear example of a conformational change switching the RIN, suitable to be visualized with RIP-MD. To do so, we analyzed MD simulations of apo-MD2 obtained

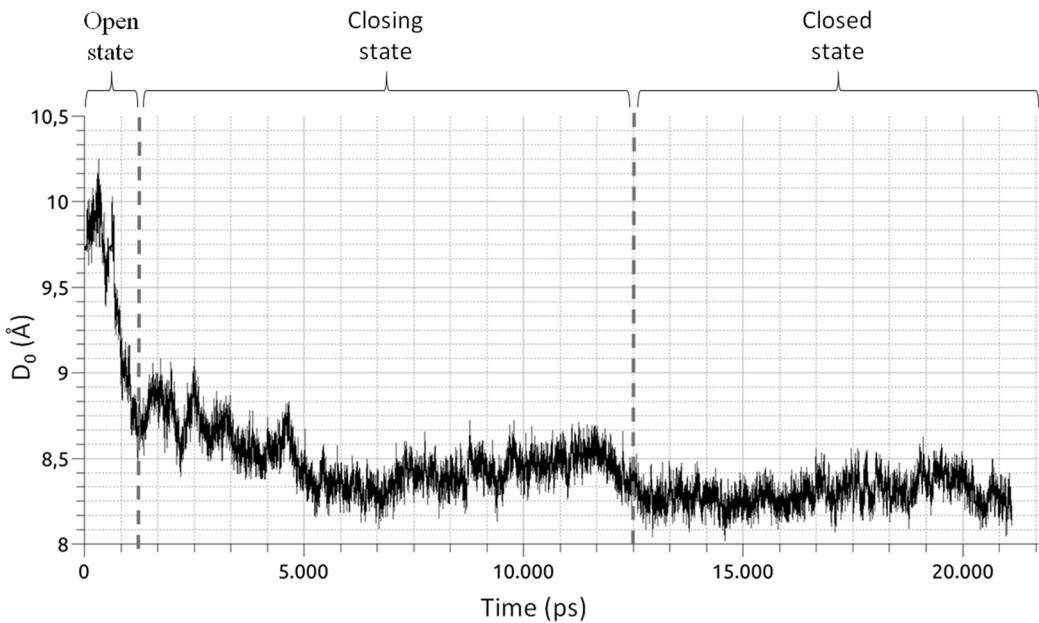

**Figure 3 Time series of the collective variable $D_0$ (Eq. (1)).** The graphic shows three distinct phases of the pocket closure, as marked by the vertical dashed lines. From 0 to 1,250 ps, the pocket is in an open conformation. From 1,250 to 12,500 ps, the closing process begins and is followed by a small opening of the pocket. From 12,500 ps until the end of the simulation, the pocket remains in a closed state.

from (*Garate & Oostenbrink, 2013*). Briefly, a 20 ns MD of a solvated MD2 was divided into three windows (Fig. 3) which characterize the closing process. The closing event along the MD simulation was projected onto the collective variable described in Eq. (1).

$$D_0 = \frac{1}{N} \sum_{n=1}^{N} \sqrt{(\mathrm{COM\beta}_n - \mathrm{COM}_{\mathrm{MD2}})^2} \qquad (1)$$

where $n$ stands for each of the 10 β strands of MD2, and $\mathrm{COM\beta}_n$ and $\mathrm{COM}_{\mathrm{MD2}}$ are the centers of mass of strand $n$ and MD2, respectively. As reported in (*Garate & Oostenbrink, 2013*), $D_0$ unambiguously differentiates the three well defined stages of the closing process. We used RIP-MD to obtain RINs for both HBs and vdW interactions between AAs, employing default parameters (see Table 1 and Supplementary Material). For each window, Pearson correlations were calculated for the number of HBs and vdW interactions for any given pair of AAs.

Residue interaction networks obtained during the closing process of MD2 (Fig. 3) for both HBs and vdW interactions between AAs, reflecting absolute Pearson correlations values $|r| \geq 0.5$ are shown in Fig. 4. There is a higher amount of correlated pairs for both HBs and vdW interactions at the initial stage, as the closing process continues the number of correlated pairs decay (see Fig. 4). The latter reveals an initial concerted action that triggers the closing event, but is lost after closure, a clear indication of the unspecific nature of non-polar interactions which are dominant within the hydrophobic cavity of MD2. To further quantify these changes, Table 2 presents the number of

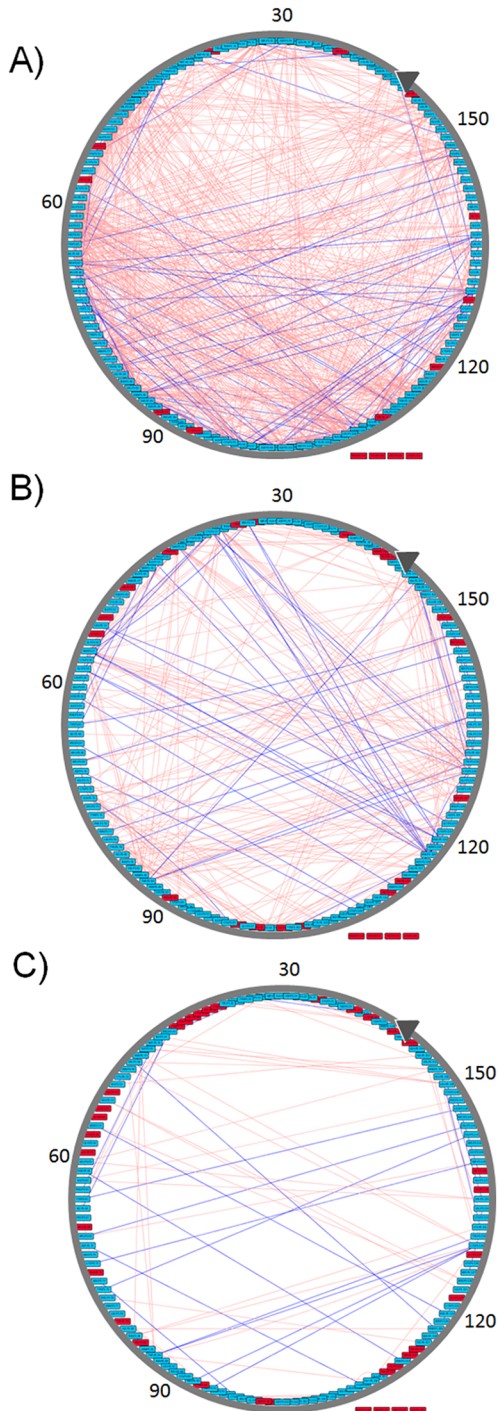

**Figure 4 Graphic representation of changes occurring in the RIN of MD2 during the three windows of its closing process.** All edges represent Pearson absolute correlation values where $|r| \geq 0.5$ in the open (A), closing (B), and closed conformation (C). Pink edges connect those AA interacting through vdW contacts while blue edges connect those AAs forming H-bonds. Red nodes indicate residues with no $|r| \geq 0.5$, nodes outside the circle, Pro50, Met85, Lys125, and Pro142, do not form any interaction $|r| \geq 0.5$ in any of the three conformational states. Images were created after loading the resulting networks in Cytoscape with circular layout sorted according to AA numbering, first AA is indicated by the gray arrow.

**Table 2 Rate of correlated pairs over the closing process of MD2 according to their absolute Pearson correlation values over the three simulation windows: open, closing, and closed conformations.**

|     | Window  | ≥0.5       | ≥0.6       | ≥0.7      | ≥0.8      | ≥0.9      |
|-----|---------|------------|------------|-----------|-----------|-----------|
| HBs | Open    | 1 (98)     | 1 (53)     | 1 (24)    | 1 (10)    | 1 (4)     |
|     | Closing | 0.64 (63)  | 0.55 (29)  | 0.54 (13) | 0.7 (7)   | 0.25 (1)  |
|     | Closed  | 0.38 (37)  | 0.40 (21)  | 0.54 (13) | 0.90 (9)  | 1.25 (5)  |
| vdW | Open    | 1 (656)    | 1 (295)    | 1 (120)   | 1 (20)    | * (0)     |
|     | Closing | 0.39 (257) | 0.35 (103) | 0.19 (23) | 0.10 (2)  | * (1)     |
|     | Closed  | 0.12 (78)  | 0.08 (23)  | 0.07 (8)  | 0 (0)     | * (0)     |

Notes:
Numbers indicate the rate between the number of correlated pairs of AAs found in the first window (open state) an the number of correlated pairs in each of the other two windows with respect to the open state for HBs and vdW interactions. In parenthesis the number of pairs of interacting AAs for each absolute Pearson correlation values.
* Indicates the absence of an interaction in the first window so it cannot be computed for the other windows.

correlated pairs at each window normalized by the amount of correlated pairs of the first window. Overall, a monotonic decrease upon window increment, not withstanding any correlation threshold, is observed. Slight deviations from the latter regarding HBs occur at Pearson values equal or above 0.7 (third and fourth columns of Table 2). This increase is due to the formation of stable H-bonds upon MD2 closure (see Table 2). On the other hand, the number of vdW interactions in Table 2 is higher than that of H-bonds, as the latter are specific interactions depending on both the relative distance and orientations of the participant residues. Finally, while most of the pairs of residues are uncorrelated, the β strands tend to be coordinated with respect to the formation and break of both types of interactions, showing how all secondary structure elements are kept in the closure of MD2.

Interestingly, this behavior is not expected for other conformational changes such as folding, in which the formation of polar interactions will lead to the appearance of highly correlated pairs, for example, the formation of an alpha-helix. In this way, RIP-MD served to quantify and visualize the counter-intuitive idea that a large conformational change can indeed lead to lower correlations, a consequence of the nature of the interactions that dominate a given structural process; in this case, non-polar VdWs interactions among residues within the MD2 cavity.

## Gap-junction channel: an example for "inter-monomeric" and "inter-molecular" interactions in a large system

GJCs are intercellular hydrophilic channels connecting the cytoplasm of two adjacent cells (*Villanelo et al., 2017*). GJCs allow the exchange of water, ions and small molecules of up to 1 kDa (*Söhl & Willecke, 2004*; *Araya-Secchi et al., 2014*). GJCs are formed by the extracellular docking of two HCs (Fig. 2C), where each HC is formed by six CXs monomers (Fig. 2D). In this example, we studied a 20 ns all-atom MD simulation of a complete GJC formed by the human CX26 (*Escalona et al., 2016*). This molecular system contains 12 identical CX monomers in total, each one comprising 226 AAs. Due to the high computational cost of studying vdW and Coulombic interactions in such a large system, these were disregarded in this example. In addition, due to the covalent nature of

**Table 3  Interactions present in different time intervals over the 20 ns GJC MD simulation.**

|            | (75%, 80%) | (80%, 85%) | (85%, 90%) | (90%, 95%) | (95%, 100%) | Total  |
|------------|-----------|-----------|-----------|-----------|------------|--------|
| $C_\alpha s$   | 133       | 169       | 173       | 239       | 10,311     | 11,025 |
| HBs        | 112       | 135       | 131       | 205       | 1,557      | 2,140  |
| SBs        | 20        | 10        | 25        | 39        | 448        | 542    |
| $\pi–\pi$   | 9         | 11        | 15        | 21        | 84         | 140    |
| Arg–Arg    | 0         | 0         | 0         | 1         | 0          | 1      |
| All        | 274       | 325       | 344       | 505       | 12,400     | 13,848 |

**Table 4  Number and type of interactions present in the human CX26 GJC.**

|         |            | HC1        |             | HC2        |             | Inter-HC |
|---------|------------|------------|-------------|------------|-------------|----------|
|         |            | Intra-chain | Inter-chain | Intra-chain | Inter-chain |          |
| Static  | HBs        | 1,312      | 101         | 1,301      | 106         | 9        |
|         | SBs        | 168        | 106         | 174        | 104         | 0        |
|         | Cation-$\pi$ | 3          | 0           | 6          | 0           | 0        |
|         | $\pi–\pi$   | 70         | 11          | 65         | 10          | 0        |
|         | Arg–Arg    | 0          | 4           | 0          | 5           | 0        |
|         | Total      | 1,553      | 222         | 1,546      | 225         | 9        |
| Dynamic | HBs        | 991        | 66          | 990        | 67          | 26       |
|         | SBs        | 183        | 101         | 170        | 88          | 0        |
|         | $\pi–\pi$   | 66         | 6           | 64         | 4           | 0        |
|         | Arg–Arg    | 0          | 0           | 0          | 1           | 0        |
|         | Total      | 1,240      | 173         | 1,224      | 160         | 26       |

**Note:**
Interactions are divided into intra-chain, inter-chain in the same HC and inter-HC monomers. The top section displays the interactions in the dynamic network (interactions present in at least 75% of the simulation), and the bottom in the network derived from the static structure.

disulfide bridges these were not considered in the following analyses. RIP-MD was run with default parameters, keeping only those appearing in at least 75% of the simulation time.

Table 3 exhibits the interactions found by RIP-MD over the human CX26 GJC simulation. All interaction types are found to be highly stable over the simulation with an overall persistence of 95–100%, indicating that this molecular structure remains structurally stable at least over the 20 ns of the MD simulation. Since $C_\alpha$ contacts describe spatial relationships between AAs with a plenty of available methods to study their interaction (Sethi et al., 2009), only the other types of interactions calculated by RIP-MD are further analyzed.

Table 4 shows those interactions occurring at least during the 75% of the MD simulation for all the following cases: interactions between AAs of the same chain for each of the two HCs (Intra-chain), between chains of the same HC (Inter-chain) and between AAs of different HCs (Inter-HCs). It should be noted that HBs and SBs are interactions appearing mainly between different chains, while $\pi–\pi$ and Arg–Arg interactions are both intra-monomeric interactions. Interestingly, only HBs appear at the interface of the two docked CXs (Fig. 5). The role of these non-bonded interactions in the

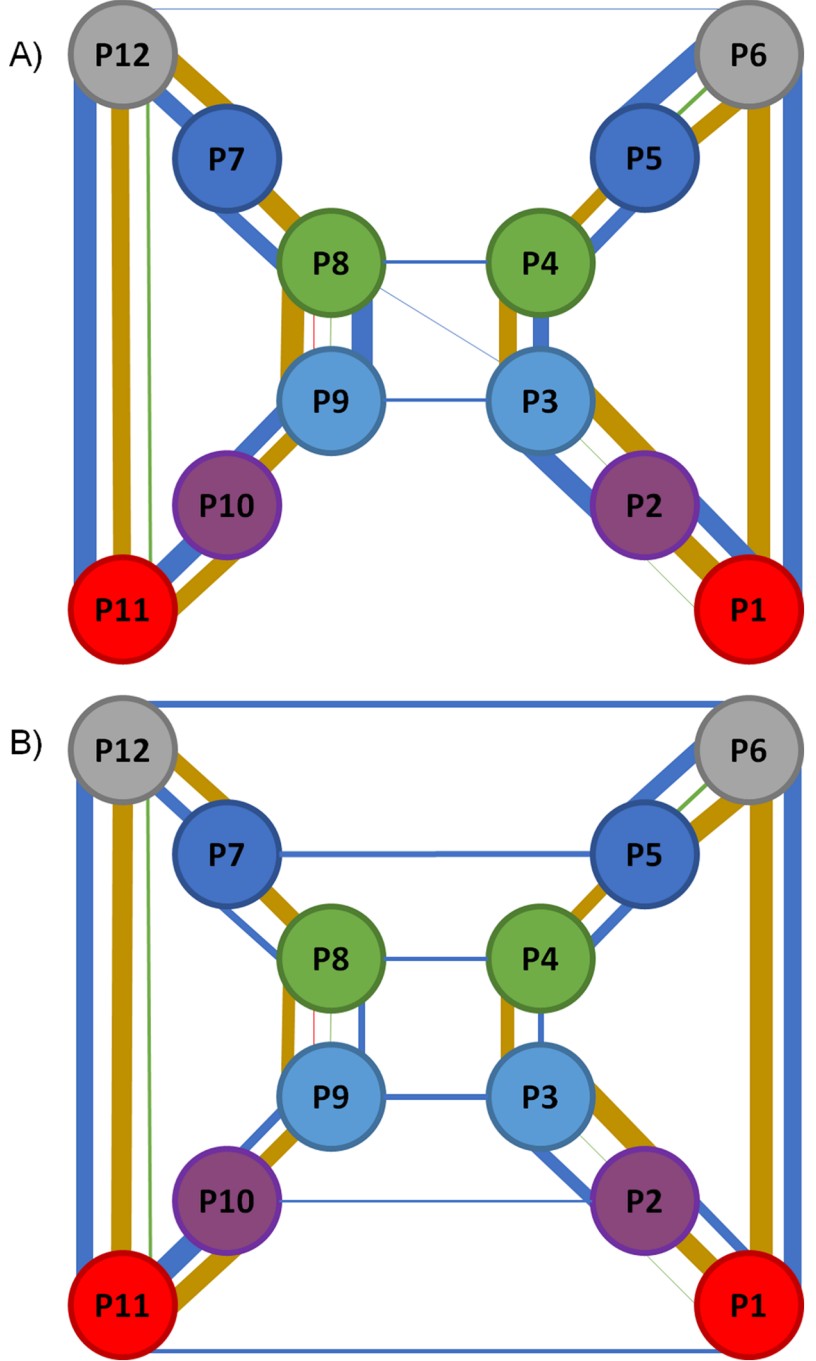

**Figure 5 Representation of the RIN formed for the static and the dynamic structure of GJC.** (A) shows the network for the static structure and (B) shows the network for the MD simulation. Each circle represents a CX subunit using a color code for subunit in each of them: red for chain A (segments P1 and P11); purple for chain (segments P2 and P10); light blue for chain C (segments P3 and P9); green for chain D (segments P4 and P8); blue for chain E (segments P5 and P7); and gray for chain F (segments P6 and P12). Interactions thickness represent the quantity of interactions, colored with the following color code: red for Arg–Arg interactions; blue for HBs; gold for SBs; and green for π–π interactions. No self-interactions are represented.

**Table 5 Number of interactions of each type appearing only in the static, in the MD simulation and in both RINs of the human CX26 GJC.**

|  |  | HBs | SBs | Cation-π | π–π | Arg–Arg |
|---|---|---|---|---|---|---|
| Static | Intra-chain | 948 | 89 | 9 | 43 | 0 |
|  | Inter-chain | 136 | 44 | 0 | 14 | 9 |
|  | Inter-HC | 7 | 0 | 0 | 0 | 0 |
| MD | Intra-chain | 316 | 100 | 0 | 38 | 0 |
|  | Inter-chain | 62 | 23 | 0 | 3 | 1 |
|  | Inter-HC | 24 | 0 | 0 | 0 | 0 |
| Both | Intra-chain | 1,665 | 253 | 0 | 92 | 0 |
|  | Inter-chain | 71 | 166 | 0 | 7 | 0 |
|  | Inter-HC | 2 | 0 | 0 | 0 | 0 |

**Note:**
Interactions are separated by type and into intra-chain. Inter-chain and inter-HCs.

maintenance of the GJC quaternary structure were also described in the original paper reporting the CX26 crystal structure (*Maeda et al., 2009*).

The information shown in the bottom section of Table 4 was then compared with the original static structure of the GJC after energy minimization (model based on PDB ID 2ZW3, see *Araya-Secchi et al. (2014)* for details). The top section of Table 4 resumes the number of interactions present in this structure. The first thing to notice when comparing the dynamic and static versions of the network is the lower number of interactions in the dynamic network. This reduction can be due to their low stability at physiological temperatures, the replacement of intra-H-bonds with H-bonds established with solvent molecules or a poor description of a given interaction by the force-field, for example, π–π interactions. Notoriously, the number inter-CXs HB interactions increases from 9 in the static structure to 26 in the dynamic network, highlighting the relevance of HBs to maintain the GJC complex. It is also very relevant that most of the inter-CXs HBs observed in the non-dynamic RIN are absent in the dynamic version, with only two HBs appearing in the entire simulation (Table 5). Regarding the other types of interactions, most of the intra and inter-chain HBs, SBs, Cation-π, and π–π interactions appear in the dynamic and non-dynamic RINs.

## CONCLUSION

Molecular dynamics simulation is a routinely employed technique to study the dynamic behavior of a system; that is, a protein. Therefore, tools that can extract relevant information in a simple and user-friendly way are urgently needed. Here, we describe RIP-MD, a method that using graph theory approaches generates RINs for different types of electrostatic interactions in protein MD simulations. We are currently working on an improved version of RIP-MD to consider waters, ions and other non-AAs molecules to generate RINs. Using RIP-MD we were able to study MD simulations of two systems: MD2 and a GJC. Regarding MD2, we focused on the study of the protein movement as reflected by Pearson correlation plots of HBs and vdW contacts. This analysis showed notable differences between the different stages of the conformational change of the

protein, revealing an initial concerted action at the beginning of the closing process, with an overall reduction of correlations for the closed state. In the case study of GJC, a comparison of the initial structure and a short MD simulation revealed that inter-chain interfaces are stabilized mainly by HBs and SBs, and that Arg–Arg and Cation-π interactions tend to disappear over the trajectory.

Residue interaction networks in protein molecular dynamics is freely available for the academic community, and it is distributed in three forms: a webserver, where users can analyze a single PDB; an standalone version that can take advantage of multi-core systems to generate these RINs; and a VMD plugin that executes the standalone version of the software and at the same time benefits from the graphical viewer of VMD. All these distributions, together with manuals and help files can be accessed from http://dlab.cl/ripmd.

## ACKNOWLEDGEMENTS

The authors would like to acknowledge F. Villanelo and other DLab members for their useful comments, suggestions and discussions of the work presented here.

### Funding

This work was partially supported by Programa de Apoyo a Centros con Financiamiento Basal AFB 17004 to Fundación Ciencia Vida; ICM-Economia project to Instituto Milenio Centro Interdisciplinario de Neurociencias de Valparaiso (CINV) [P09-022-F]; FONDECYT projects [1160574, 11140342, 1181089]; from the US Air Force Office of Scientific Research [FA9550-16-1-0384]; and Beca de Asistencia Academica from Universidad Nacional Andres Bello to Sebastian Contreras-Riquelme. This research was also supported by the supercomputing infrastructure of the Chilean National Laboratory for High Performance Computing (NLHPC) [ECM-02]. There was no additional external funding received for this study. The funders had no role in study design, data collection and analysis, decision to publish, or preparation of the manuscript.

### Grant Disclosures

The following grant information was disclosed by the authors:
Programa de Apoyo a Centros con Financiamiento Basal AFB 17004 to Fundación Ciencia Vida.
ICM-Economia project to Instituto Milenio Centro Interdisciplinario de Neurociencias de Valparaiso (CINV): P09-022-F.
FONDECYT projects: 1160574, 11140342, 1181089.
US Air Force Office of Scientific Research: FA9550-16-1-0384.
Beca de Asistencia Academica from Universidad Nacional Andres Bello to Sebastian Contreras-Riquelme.
Chilean National Laboratory for High-Performance Computing (NLHPC): ECM-02.

## Competing Interests

Tomas Perez-Acle is an Academic Editor for PeerJ.

## Author Contributions

- Sebastián Contreras-Riquelme performed the experiments, analyzed the data, contributed reagents/materials/analysis tools, prepared figures and/or tables, authored or reviewed drafts of the paper, approved the final draft.
- Jose-Antonio Garate conceived and designed the experiments, analyzed the data, authored or reviewed drafts of the paper, approved the final draft.
- Tomas Perez-Acle contributed reagents/materials/analysis tools, approved the final draft, he provided lab space and computing equipment.
- Alberto J.M. Martin conceived and designed the experiments, analyzed the data, contributed reagents/materials/analysis tools, prepared figures and/or tables, authored or reviewed drafts of the paper, approved the final draft.

## Data Availability

The RIP-MD Code can be obtained at the RIP-MD webserver located at http://dlab.cl/ripmd in the "Download RIP-MD" tab. The whole project can also be obtained at the RIP-MD GitHub.

GitHub: https://github.com/DLab/RIP-MD.

Supplementary data for MD2 and GJC (Trajectories, models, RIP-MD's results and GJC pearson networks as cytoscape sessions) is also available at the RIP-MD webserver (http://dlab.cl/ripmd) under the "Supp. Data" tab.

## Supplemental Information

Supplemental information for this article can be found online at http://dx.doi.org/10.7717/peerj.5998#supplemental-information.

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
