# Peer review of "RIP-MD: a tool to study residue interaction networks in protein molecular dynamics"

_PeerJ, doi:10.7717/peerj.5998_

## Round 0.1 · original submission · Major Revisions

After getting the three detailed reviews, and after re-inspecting the manuscript for a second time, considering rejection of the manuscript, I finally camed the conclusion to give you a chance to drastically improve the manuscript for a major revision, which should follow strictly the recommendations of especially reviewer 3, but also respond in detail to the very strong criticism expressed by reviewer 1, who finally came to the conclusion that " No benchmarking and comparison is made with the performance of other RIP codes. Authors missed also the opportunity to embed their analysis results into previous research findings to support the importance of the use of their RIP-MD code. If there is no such result for these two systems, other system should have been analyzed." I agree very much with the benchmarking remark, and actually statistics was mentioned also by reviewer 3.

In addition, the paper presents the software as an easy-to-use tool for advanced structural protein analysis but the actual software problems that two of the reviewers report should be taken very serious and fixed, as the present state seems to be more a beta-version than a real mature software package.

In summery, only a very extensive revision addressing all above mentioned points to the reviewers satisfaction can be re-considered for publication, but I would also on the other hand encourage the authors to make this attempt and undergo this effort as their tool might indeed be potentially useful.

Reviewer 1 ·

Basic reporting

The manuscript describes an analysis tool for interaction network of protein static and dynamic structures. First part of the manuscript is more like a program documentation, which attempts to show some features of the script package based on MDAnalysis. After the program description, authors showed results of their RIN analysis for two case studies.
General comments are as follows:
(1) It is recommended the authors find a native English speaker to proofread the manuscript. I believe some of their main findings is not recognized by the reviewer due to unclear English language used. Please don’t use ‘van der Waalss’
(2) Most of the figures need improvement. For example, it took some time to find “i” and “e” in Figure 2 (According to figure caption, they stand for intra- and extracellular spaces). While Figure 5 is matured and sophisticated, Figure 4 is far less informative.
(3) Notations are unclear and not systematic in the text.

Experimental design

To my opinion, the original aim of the manuscript was to provide an easy-to-use tool for advanced structural protein analysis and to demonstrate usefulness of the RIP-MD with two examples. Authors missed this aim, since it remained unclear what kind of extra chemical knowledge can be achieved by using RIP-MD which cannot be obtained by a few lines of scripting code. In the beginning of Section 4.2, authors stated that RIP analysis of vdW and Coulombic interactions are “disregarded” showing limitation of the current implementation, however there is nothing said about other limitations. RIP analysis neglects the protein interactions with the surrounding molecules (e.g. membrane molecules, water as well as ions), however these species play an essential role in the protein structure making difficult to draw a proper chemical conclusion about the system studied from RIP analysis. Although, RIP analysis would be an important part of a more general interaction analysis tool. Further, the estimation of the half-lifetime for interacting residues would be also very useful to identify the important interaction sites.
The chemistry part of the methodology can be found only in Supplementary Information and therefore Methods section became very technical and superficial. A short definition of the interaction used in the analysis should have been the part of the main text.
Selection of the two case studies is not obvious, explanation should have been given.

Validity of the findings

Besides issues discussed above, I think, the manuscript is lack of novelty in the present form. No benchmarking and comparison is made with the performance of other RIP codes. Authors missed also the opportunity to embed their analysis results into previous research findings to support the importance of the use of their RIP-MD code. If there is no such result for these two systems, other system should have been analyzed.

·

Basic reporting

The methods are described properly and with sufficient details. Illustration of the method on a sample protein is very useful. English is, as far as I can judge, good. Literature is properly cited.

Experimental design

The methods described in the article are not completely new but they are placed into a nice web program and nice VMD plug in. It fits well the journal.

Validity of the findings

I tried to run the server and to install and run the VMD plug in (in the PDB mode only, not for trajectory) and everything was running smoothly. The only exception was that when I did some analysis (e.g. H-bonds only) and I decided to do another analysis (e.g. cation-pi) it seems that the program freezes (a hint for improvement). It works if used canonically.

Additional comments

I believe authors can fix or explain the above mentioned problem (freezing when redoing the analysis).

·

Basic reporting

The article "RIP-MD: A tool to study residue interaction networks in protein molecular dynamics" by Contreras-Riquelme et al. describe software tool RIP-MD for analysis of residue interaction networks online and as a VMD plugin in molecular dynamics.

This software tool might be interesting, but this reviewer failed to install RIP-MD core program on his freshly installed Windows 10 computer and therefore was only able to analyze the data presented within the manuscript.

The article is written in well-readable English meeting professional standards.

Figures are appropriately labeled, but some minor adjustments will be mentioned in the latest section of the review. It would be also great to put into the SI and manual how to get visualizations using Cytoscape as the results are not provided from within VMD plugin (well at least according to manual).

The article is self-contained description of piece of software and its use.

Experimental design

Submission clearly define the research question, which is relevant and meaningful - to find interaction network between residues, but as stated in previous section, reviewer was unable to install the software in full to actually use the software in all possible described ways.

Selection of testing proteins and simulation protocol is appropriate; however selection of several cutoffs is beyond this reviewer comprehension as will be stated in the author comments section.

Validity of the findings

Data provided within the manuscript are hard to follow. It is true that the visualization of all interactions within a medium sized protein of around 500 residues is quite large given approximate average of 20 interactions of all types per residue and therefore a reviewer would expect a list of filters or any protocol for meaningful pruning of the data to obtain the better overview over the effect of simulation on e.g. the subset of specified residues. The only exception of coulomb and vdw interactions, which are pruned by thermal noise.

However, this note may be hindered by the fact of only visualization which was available to the reviewer – which was visualization of the webserver generated results within the Cytoscape 3.6 or only statically within VMD plugin.

Additional comments

Here I list all troubling points which I have identified thorough manuscript, SI and manual of the program:

1. As stated above, this reviewer had access only to the freshly installed Windows 10 computer and as such he was unable to run other analysis than using webserver. While the installation of VMD plugin on Windows is well described, the installation of the core program is provided only for Linux using bash or apt-get installation scripts. Since the program is written in Python, it would be nice to provide installation instruction of core program on Windows (and possibly Mac) within manual.
2. Manual is also lacking instructive example of the RIN analysis on static structure as well as MD trajectory using both Cytoscape and VMD plugin.
3. Webserver is not providing any visualization of the results, nor at least the statistics, which is a bit pity.
4. While I have submitted several jobs with my email address, I have never received email with confirmation that the job is done.
5. RIN analysis does not consider HETATMs. This is unfortunate as the interactions of the ligands within binding pockets are of utmost importance in the drug design field.
6. Description, how to obtain figures should be given within SI as it would be instructive for future users of the program
7. Citations are a little bit tricky within the article. As an example, in sentence
„RIP-MD is available as a web server for static RIN derivation using PDB files [2], or as a Visual Molecular Dynamics (VMD) [19] plugin to obtain dynamic RINs derived from MD trajectories.“
reviewer expected that ref. [2] will be citation to web server and reference [19] will be citation to VMD plugin not to PDB database and VMD software themselves.
8. Windows in Table 1 and Figures 3 and 4 should be described as open, closing and closed cavity
9. It is not understandable, what are numbers in Table 1 before brackets.
10. While quantification of the interactions like the ones in Tables can be usable, the analysis of protein structure from MD or from single structure are often more qualitative in the nature, e.g. interactions around ligand or any specified residue, identification of the most interacting residue or identification of the correlated motions. The manuscript unfortunately fail to provide such examples.
11. The selection of the cut offs of individual parameters is not entirely understandable
a. why is distance between Calphas 8A? That will give unnecesarily large numbers of interacting pairs
b. HB parameters are ok (3A), but distance for salt bridge is unnecesarily large (6A) given the electrostatic screening by the water solvent.
12. P7 – typo in COMbetan, please check typos within whole manuscript

---

## Round 0.2 · Major Revisions

Although your revised manuscript addresses a large part of the concerns raised by the reviewers, reviewer 3 still feels that his concerns and recommendations were not adequately addressed. The extend of concerns allows only the decision "major revision" and I would like to recommend you to follow reviewer 3 suggestions in detail if you consider re-submission.

·

Basic reporting

see report for the previous version

Experimental design

see report for the previous version

Validity of the findings

see report for the previous version

Additional comments

I am happy with the changes made by authors regarding my comment and also responses to other reviewers look reasonable.

·

Basic reporting

Since this is a re-review, I think that the most useful format will be answering to remaining individual arguments in rebuttal letter:

> We also reviewed the English and made some minor changes to improve how the article reads.

Still needs work
Check of language and typos is highly recommended even in the newly added text (examples: cationπ without dash, maitaining, Figure vz figs., RIP-MP instead of RIP-MD, etc.)

> We also included a summary table describing all interactions calculated by RIP-MD but given the length of their complete description users/reader will still have to refer to the supplementary material.

Thanks for clarification in maintext

>> Citations are a little bit tricky within the article. As an example, in sentence
„RIP-MD is available as a web server for static RIN derivation using PDB files [2], or as a Visual Molecular Dynamics (VMD) [19] plugin to obtain dynamic RINs derived from MD trajectories.“
reviewer expected that ref. [2] will be citation to web server and reference [19] will be citation to VMD plugin not to PDB database and VMD software themselves.

> We improved this and several other minors issued in the text.

Improve them more - References are not properly formatted nor consistent to each other - e.g. ref. 3, 5, 6, 7, 8, 9, ...

>> It is not understandable, what are numbers in Table 1 before brackets.

> This is better explained now, as you will notice in the text.

Increase the size of font in Table 1

Experimental design

> First, RIP-MD only works in Unix systems, we know it can be made to work on Mac OS but it was designed and tested on Ubuntu and Debian OS. This incompatibility is caused by some of the libraries we employed to build it being only available for Unix systems (MDAnalysis). To overcome this incompatibility, we made available a virtual machine with RIP-MD pre-installed under an Ubuntu distribution, so it can be used under Windows and Mac OS with very little effort. One should keep in mind that VMD is licensed so we can not distribute it in the same virtual machine.

OK, but it should be explained on the webpage.

>> It would be also great to put into the SI and manual how to get visualizations using Cytoscape as the results are not provided from within VMD plugin (well at least according to manual).

> We now provide as supplementary material all scripts employed to generate the figures shown in the article as well as improved versions of the SI and manual file where readers can find a step by step guide to analyze RINs in Cytoscape.

I fail to see any scripts in SI.
Manual should be also added to SI
The description of the analysis in Cytoscape do not cover the visualization of the example result in Cytoscape window and what to see from the presented data -> e.g. which values are important for what.

>> Webserver is not providing any visualization of the results, nor at least the statistics, which is a bit pity.

>The web server is now providing a summary table showing the occurrence of each type of interaction in the query PDB file. For proper visualization of the resulting RIN user must employ Cytoscape or VMD (see user manual where a detailed explanation on how to do this is provided).

If the results of the analysis would be provided directly on web (e.g. listing of the most important contacts + some visualization of the contacts in the structure e.g. JSmol, LiteMol, NGLviewer) In that case, any user might consider to try the tool on static structure online first to see what can be learned from those results before pursuing the installation of software itself and third party software as well (especially for Cytoscape, whose use for analysis is not entirely easy for first time user - I frankly still do not know what I should be specifically doing after few hours spent on this review and typical user will not have such patience.)

Validity of the findings

> In second place and with respect to the benchmarking, we do not know about any other software that allows to create multigraphs from MD trajectories so we could not perform it. The only possibility is to compare it with the method presented on reference 33, which only considers Calpha contacts making us disregard this option, or to employ other software that generates RINs from static PDB structures, something that is exactly what RIP-MD aims to prevent.

Why haven’t you provided comparison at least for Calpha contacts then?
I have found multiple downloadable software that allows analysis of the contact maps on trajectories against which it would be interesting to run benchmarking of results on the RIP-MD -> MDcons, ConAn, PROTMAP2D, MD-TASK, MDAnalysis.org …

> What we did instead was to report the computational resources and times we employed to carry out the analysis described in the article in each of its steps.

Time is not that important as the benchmarking of the validity of the findings.

>> It remained unclear what kind of extra chemical knowledge can be achieved by using RIP-MD which cannot be obtained by a few lines of scripting code. In the beginning of Section 4.2, authors stated that RIP analysis of vdW and Coulombic interactions are “disregarded” showing limitation of the current implementation, however there is nothing said about other limitations. RIP analysis neglects the protein interactions with the surrounding molecules (e.g. membrane molecules, water as well as ions), however these species play an essential role in the protein structure making difficult to draw a proper chemical conclusion about the system studied from RIP analysis. Although, RIP analysis would be an important part of a more general interaction analysis tool. Further, the estimation of the half-lifetime for interacting residues would be also very useful to identify the important interaction sites.

> RIP-MD is meant to be an intuitive and easy-to-follow visualization tool for (pairwise) residue interactions, the latter is particularly relevant for MD in which the generated data is enormous. Consequently, we do not expect to produce new chemical data (the physico-chemical information is already present in the phase-space trajectories); however, due to the employment of networks, the users can take advantage of the mathematical formalism of network theory (i.e. by using Cytoscape) to further analyze their simulations. The main novelty of our tool is that it adds many more descriptors than contact maps and includes the capacity for analyzing MD trajectories, a sentence in the manuscript emphasizing the features of our tools was added in the main manuscript.

Yes, physico-chemical information is already present in the phase space trajectories, but RIP-MD fails to provide simple measures to mine this information for easy comprehension of the user of the software let alone further use in analysis and presentation of research findings towards (scientific) public.

Nothing is however still said about limitations of RIP analysis – that it neglects the protein interactions with the surrounding molecules (e.g. membrane molecules, water as well as ions), however these species play an essential role in the protein structure making difficult to draw a proper chemical conclusion about the system studied from RIP analysis.

>>Selection of the two case studies is not obvious, explanation should have been given.
Besides issues discussed above, I think, the manuscript is lack of novelty in the present form. No benchmarking and comparison is made with the performance of other RIP codes. Authors missed also the opportunity to embed their analysis results into previous research findings to support the importance of the use of their RIP-MD code. If there is no such result for these two systems, other system should have been analyzed.

> Regarding the two examples present in the main manuscript; i) The first example (Fig. 4) is meant to show how RIP-MD can visually discriminate among the different stages of a conformational change and how correlated interactions guide the process and are lost when the "closed" state is reached, which might be a common trend in process guided by non-polar interactions; this is indeed "new" chemical information that was not reported in Ref. 14. (the original studies in which for the first time the closing process was reported); ii) The second example is meant to emphasize how interactions in x-ray protein crystals are not necessarily the same. Moreover, the visualization power of RIP-MD is highlighted as a very complex system (12 monomers) is very easily visualized by our method. A sentence in the main manuscript was added, emphasizing the previous.

The explanation of the selection of given examples was added in sufficient detail, however the results of the analysis of the second system are not explained well.

Table 5 contains several weird results – last lines for Both combining Dynamic and Static results (or maybe I am reading wrongly its description) should have at least as contacts as the maximal value in the previous lines. This part requires further clarification.

Additional comments

>> Most of the figures need improvement. For example, it took some time to find “i” and “e” in Figure 2 (According to figure caption, they stand for intra- and extracellular spaces). While Figure 5 is matured and sophisticated, Figure 4 is far less informative.

> We also corrected all these, figure 2 has a larger font now to indicate the different cellular compartments and improved the detail of figure 4 by using color codes to differentiate between interactions and amino acids with and without interactions.

Please enlarge font in Figure 1 and Figure 5
Are all edges in the Figure 4 similarly distributed? Can you indicate where is residue 1, 100, 200, etc for clarification?
Table 3 is over the edge.

>> Description, how to obtain figures should be given within SI as it would be instructive for future users of the program

> We now provide as supplementary material all scripts employed to generate the figures shown in the article as well as an improved version of the SI where readers can find a step by step guide to analyze RINs in Cytoscape.

I have seen no scripts in Supplemental Informations.

>> While quantification of the interactions like the ones in Tables can be usable, the analysis of protein structure from MD or from single structure are often more qualitative in the nature, e.g. interactions around ligand or any specified residue, identification of the most interacting residue or identification of the correlated motions. The manuscript unfortunately fail to provide such examples.

> Further analysis as those commented by the reviewer can be easily carried out within Cytoscape, where by adding further information to the RINs (e.g. residue conservation) as node attributes will help in the processes. Proper instructions to perform such analysis are now provided in both the manual as a work example.

The explanation of usage of Cytoscape is still insufficient.

---

## Round 0.3 · accepted · Accept

Thank you for your careful revision, I consider all remarks and concerns addressed properly and to our full satisfaction.

#